# Bayesian inference for low rank spatiotemporal neural receptive fields

**Mijung Park**
Electrical and Computer Engineering
The University of Texas at Austin
mjpark@mail.utexas.edu

**Jonathan W. Pillow**
Center for Perceptual Systems
The University of Texas at Austin
pillow@mail.utexas.edu

## Abstract

The receptive field (RF) of a sensory neuron describes how the neuron integrates sensory stimuli over time and space. In typical experiments with naturalistic or flickering spatiotemporal stimuli, RFs are very high-dimensional, due to the large number of coefficients needed to specify an integration profile across time and space. Estimating these coefficients from small amounts of data poses a variety of challenging statistical and computational problems. Here we address these challenges by developing Bayesian reduced rank regression methods for RF estimation. This corresponds to modeling the RF as a sum of space-time separable (i.e., rank-1) filters. This approach substantially reduces the number of parameters needed to specify the RF, from 1K-10K down to mere 100s in the examples we consider, and confers substantial benefits in statistical power and computational efficiency. We introduce a novel prior over low-rank RFs using the restriction of a matrix normal prior to the manifold of low-rank matrices, and use "localized" row and column covariances to obtain sparse, smooth, localized estimates of the spatial and temporal RF components. We develop two methods for inference in the resulting hierarchical model: (1) a fully Bayesian method using blocked-Gibbs sampling; and (2) a fast, approximate method that employs alternating ascent of conditional marginal likelihoods. We develop these methods for Gaussian and Poisson noise models, and show that low-rank estimates substantially outperform full rank estimates using neural data from retina and V1.

## 1 Introduction

A neuron's linear receptive field (RF) is a filter that maps high-dimensional sensory stimuli to a one-dimensional variable underlying the neuron's spike rate. In white noise or reverse-correlation experiments, the dimensionality of the RF is determined by the number of stimulus elements in the spatiotemporal window influencing a neuron's probability of spiking. For a stimulus movie with $n_x \times n_y$ pixels per frame, the RF has $n_x n_y n_t$ coefficients, where $n_t$ is the (experimenter-determined) number of movie frames in the neuron's temporal integration window. In typical neurophysiology experiments, this can result in RFs with hundreds to thousands of parameters, meaning we can think of the RF as a vector in a very high dimensional space.

In high dimensional settings, traditional RF estimators like the whitened spike-triggered average (STA) exhibit large errors, particularly with naturalistic or correlated stimuli. A substantial literature has therefore focused on methods for regularizing RF estimates to improve accuracy in the face of limited experimental data. The Bayesian approach to regularization involves specifying a prior distribution that assigns higher probability to RFs with particular kinds of structure. Popular methods have involved priors to impose smallness, sparsity, smoothness, and localized structure in RF coefficients[1, 2, 3, 4, 5].

Here we develop a novel regularization method to exploit the fact that neural RFs can be modeled as a low-rank matrices (or tensors). This approach is justified by the observation that RFs can be well described by summing a small number of space-time separable filters [6, 7, 8, 9]. Moreover, it can substantially reduce the number of RF parameters: a rank $p$ receptive field in $n_x n_y n_t$ dimensions requires only $p(n_x n_y + n_t - 1)$ parameters, since a single space-time separable filter has $n_x n_y$ spatial coefficients and $n_t - 1$ temporal coefficients (i.e., for a temporal unit vector). When $p \ll \min(n_x n_y, n_t)$, as commonly occurs in experimental settings, this parametrization yields considerable savings.

In the statistics literature, the problem of estimating a low-rank matrix of regression coefficients is known as *reduced rank regression* [10, 11]. This problem has received considerable attention in the econometrics literature, but Bayesian formulations have tended to focus on non-informative or minimally informative priors [12]. Here we formulate a novel prior for reduced rank regression using a restriction of the matrix normal distribution [13] to the manifold of low-rank matrices. This results in a marginally Gaussian prior over RF coefficients, which puts it on equal footing with "ridge", AR1, and other Gaussian priors. Moreover, under a linear-Gaussian response model, the posterior over RF rows and columns are conditionally Gaussian, leading to fast and efficient sampling-based inference methods. We use a "localized" form for the row and and column covariances in the matrix normal prior, which have hyperparameters governing smoothness and locality of RF components in space and time [5]. In addition to fully Bayesian sampling-based inference, we develop a fast approximate inference method using coordinate ascent of the conditional marginal likelihoods for temporal (column) and spatial (row) hyperparameters. We apply this method under linear-Gaussian and linear-nonlinear-Poisson encoding models, and show that the latter gives the best performance on neural data.

The paper is organized as follows. In Sec. 2, we describe the low-rank RF model with localized priors. In Sec. 3, we describe a fully Bayesian inference method using the blocked-Gibbs sampling with interleaved Metroplis Hastings steps. In Sec. 4, we introduce a fast method for approximate inference using conditional empirical Bayesian hyperparameter estimates. In Sec. 5, we extend our estimator to the linear-nonlinear Poisson encoding model. Finally, in Sec. 6, we show applications to simulated and real neural datasets from retina and V1.

## 2 Hierarchical low-rank receptive field model

### 2.1 Response model (likelihood)

We begin by defining two probabilistic encoding models that will provide likelihood functions for RF inference. Let $y_i$ denote the number of spikes that occur in response to a $(d_t \times d_x)$ matrix stimulus $X_i$, where $d_t$ and $d_x$ denote the number of temporal and spatial elements in the RF, respectively. Let $K$ denote the neuron's $(d_t \times d_x)$ matrix receptive field.

We will consider, first, a linear Gaussian encoding model:

$$y_i | X_i \quad \sim \quad \mathcal{N}(\mathbf{x}_i^\top \mathbf{k} + b, \gamma), \tag{1}$$

where $\mathbf{x}_i = \text{vec}(X_i)$ and $\mathbf{k} = \text{vec}(K)$ denote the vectorized stimulus and vectorized RF, respectively, $\gamma$ is the variance of the response noise, and $b$ is a bias term. Second, we will consider a linear-nonlinear-Poisson (LNP) encoding model

$$y_i | X_i, \quad \sim \quad \text{Poiss}(g(\mathbf{x}_i^\top \mathbf{k} + b)). \tag{2}$$

where $g$ denotes the nonlinearity. Examples of $g$ include exponential and soft rectifying function, $\log(\exp(\cdot) + 1)$, both of which give rise to a concave log-likelihood [14].

### 2.2 Prior for low rank receptive field

We can represent an RF of rank $p$ using the factorization

$$K \quad = \quad K_t K_x^\top, \tag{3}$$

where the columns of the matrix $K_t \in R^{d_t \times p}$ contain temporal filters and the columns of the matrix $K_x \in R^{d_x \times p}$ contain spatial filters.

We define a prior over rank-$p$ matrices using a restriction of the matrix normal distribution $\mathcal{MN}(0, C_x, C_t)$. The prior can be written:

$$p(K|C_t, C_x) = \frac{1}{Z} \exp\left(-\frac{1}{2}\text{Tr}[C_x^{-1}K^\top C_t^{-1}K]\right), \tag{4}$$

where the normalizer $Z$ involves integration over the space of rank-$p$ matrices, which has no known closed-form expression. The prior is controlled by a "column" covariance matrix $C_t \in \mathbb{R}^{d_t \times d_t}$ and "row" covariance matrix $C_x \in \mathbb{R}^{d_x \times d_x}$, which govern the temporal and spatial RF components, respectively.

If we express $K$ in factorized form (eq. 3), we can rewrite the prior

$$p(K|C_t, C_x) = \frac{1}{Z} \exp\left(-\frac{1}{2}\text{Tr}\left[(K_x^\top C_x^{-1} K_x)(K_t^\top C_t^{-1} K_t)\right]\right). \tag{5}$$

This formulation makes it clear that we have conditionally Gaussian priors on $K_t$ and $K_x$, that is:

$$\mathbf{k}_t|\mathbf{k}_x, C_x, C_t \sim \mathcal{N}(0, A_x^{-1} \otimes C_t),$$
$$\mathbf{k}_x|\mathbf{k}_t, C_t, C_x \sim \mathcal{N}(0, A_t^{-1} \otimes C_x), \tag{6}$$

where $\otimes$ denotes Kronecker product, and $\mathbf{k}_t = \text{vec}(K_t) \in \mathbb{R}^{pd_t \times 1}$, $\mathbf{k}_x = \text{vec}(K_x) \in \mathbb{R}^{pd_x \times 1}$, and where we define $A_x = K_x^\top C_x^{-1} K_x$ and $A_t = K_t^\top C_t^{-1} K_t$.

We define $C_t$ and $C_x$ have a parametric form controlled by hyperparameters $\theta_t$ and $\theta_x$, respectively. This form is adopted from the "automatic locality determination" (ALD) prior introduced in [5]. In the ALD prior, the covariance matrix encodes the tendency for RFs to be localized in both space-time and spatiotemporal frequency.

For the spatial covariance matrix $C_x$, the hyperparameters are $\theta_x = \{\rho, \mu_s, \mu_f, \Phi_s, \Phi_f\}$, where $\rho$ is a scalar determining the overall scale of the covariance; $\mu_s$ and $\mu_f$ are length-$D$ vectors specifying the center location of the RF support in space and spatial frequency, respectively (where $D$ is the number of spatial dimensions, e.g., "D=2" for standard 2D visual pixel stimuli). The positive definite matrices $\Phi_s$ and $\Phi_f$ are $D \times D$ determine the size of the local region of RF support in space and spatial frequency, respectively [15]. In the temporal covariance matrix $C_t$, the hyperparameters $\theta_t$, which are directly are analogous to $\theta_x$, determine the localized RF structure in time and temporal frequency.

Finally, we place a zero-mean Gaussian prior on the (scalar) bias term: $b \sim \mathcal{N}(0, \sigma_b^2)$.

## 3 Posterior inference using Markov Chain Monte Carlo

For a complete dataset $\mathcal{D} = \{X, \mathbf{y}\}$, where $X \in \mathcal{R}^{n \times (d_t d_x)}$ is a design matrix, and $\mathbf{y}$ is a vector of responses, our goal is to infer the joint posterior over $K$ and $b$,

$$p(K, b|\mathcal{D}) \propto \int\int p(\mathcal{D}|K, b) p(K|\theta_t, \theta_x) p(b|\sigma_b^2) p(\theta_t, \theta_x, \sigma_b^2) d\sigma_b^2 d\theta_t d\theta_x. \tag{7}$$

We develop an efficient Markov chain Monte Carlo (MCMC) sampling method using blocked-Gibbs sampling. Blocked-Gibbs sampling is possible since the closed-form conditional priors in eq. 6 and the Gaussian likelihood yields closed-form "conditional marginal likelihood" for $\theta_t|(\mathbf{k}_x, \theta_x, D)$ and $\theta_x|(\mathbf{k}_t, \theta_t, D)$, respectively[1]. The blocked-Gibbs first samples $(\sigma_b^2, \theta_t, \gamma)$ from the conditional evidence and simultaneously sample $\mathbf{k}_t$ from the conditional posterior. Given the samples of $(\sigma_b^2, \theta_t, \gamma, b, \mathbf{k}_t)$, we then sample $\theta_x$ and $\mathbf{k}_x$ similarly.

For sampling from the conditional evidence, we use the Metropolis Hastings (MH) algorithm to sample the low dimensional space of hyperparameters. For sampling $(b, \mathbf{k}_t)$ and $\mathbf{k}_x$, we use the closed-form formula (will be introduced shortly) for the mean of the conditional posterior. The details of our algorithm are as follows.

**Step 1** Given $(i\text{-}1)$th samples of $(\mathbf{k}_x, \theta_x)$, we draw $i$th samples $(b, \mathbf{k}_t, \theta_t, \sigma_b^2, \gamma)$ from

$$p(b^{(i)}, \mathbf{k}_t^{(i)}, \theta_t^{(i)}, \sigma_b^{2(i)}, \gamma^{(i)}|\mathbf{k}_x^{(i-1)}, \theta_x^{(i-1)}, \mathcal{D}) = p(\theta_t^{(i)}, \sigma_b^{2(i)}, \gamma^{(i)}|\mathbf{k}_x^{(i-1)}, \theta_x^{(i-1)}, \mathcal{D})$$
$$p(b^{(i)}, \mathbf{k}_t^{(i)}|\theta_t^{(i)}, \sigma_b^{2(i)}, \gamma^{(i)}, \mathbf{k}_x^{(i-1)}, \theta_x^{(i-1)}, \mathcal{D}),$$

which is divided into two parts[2]:

- We sample $(\theta_t, \sigma_b^2, \gamma)$ from the conditional posterior given by

$$
\begin{aligned}
p(\theta_t, \sigma_b^2, \gamma | \mathbf{k}_x, \theta_x, \mathcal{D}) &\propto p(\theta_t, \sigma_b^2, \gamma) \int p(\mathcal{D}|b, \mathbf{k}_t, \mathbf{k}_x, \gamma) p(b, \mathbf{k}_t | \mathbf{k}_x, \theta_x, \theta_t) db d\mathbf{k}_t, \\
&\propto p(\theta_t, \sigma_b^2, \gamma) \int \mathcal{N}(\mathcal{D}|M_x'\mathbf{w}_t, \gamma I) \mathcal{N}(\mathbf{w}_t|0, C_{\mathbf{w}_t}) d\mathbf{w}_t, \quad (8)
\end{aligned}
$$

where $\mathbf{w}_t$ is a vector of $[b \; \mathbf{k}_t^T]^T$, $M_x'$ is concatenation of a vector of ones and the matrix $M_x$, which is generated by projecting each stimulus $X_i$ onto $K_x$ and then stacking it in each row, meaning that the $i$-th row of $M_x$ is $[\text{vec}(X_i K_x)]^\top$, and $C_{\mathbf{w}_t}$ is a block diagonal matrix whose diagonal is $\sigma_b^2$ and $A_x^{-1} \otimes C_t$. Using the standard formula for a product of two Gaussians, we obtain the closed form conditional evidence:

$$
p(\mathcal{D}|\theta_t, \sigma_b^2, \gamma, \mathbf{k}_x, \theta_x) \approx \frac{|2\pi\Lambda_t|^{\frac{1}{2}}}{|2\pi\gamma I|^{\frac{1}{2}}|2\pi C_{\mathbf{w}_t}|^{\frac{1}{2}}} \exp\left[\frac{1}{2}\boldsymbol{\mu}_t^\top \Lambda_t^{-1} \boldsymbol{\mu}_t - \frac{1}{2\gamma}\mathbf{y}^\top \mathbf{y}\right] \quad (9)
$$

where the mean and covariance of conditional posterior over $\mathbf{w}_t$ given $\mathbf{k}_x$ are given by

$$
\boldsymbol{\mu}_t = \frac{1}{\gamma}\Lambda_t M_x'^T \mathbf{y}, \quad \text{and} \quad \Lambda_t = (C_{\mathbf{w}_t}^{-1} + \frac{1}{\gamma}M_x'^T M_x)^{-1}. \quad (10)
$$

We use the MH algorithm to search over the low dimensional hyperparameter space, with the conditional evidence (eq. 9) as the target distribution, under a uniform hyperprior on $(\theta_t, \sigma_b^2, \gamma)$.

- We sample $(b, \mathbf{k}_t)$ from the conditional posterior given in eq. 10.

**Step 2** Given the $i$th samples of $(b, \mathbf{k}_t, \theta_t, \sigma_b^2, \gamma)$, we draw $i$th samples $(\mathbf{k}_x, \theta_x)$ from

$$
\begin{aligned}
p(\mathbf{k}_x^{(i)}, \theta_x^{(i)} | b^{(i)}, \mathbf{k}_t^{(i)}, \sigma_b^{2(i)}, \theta_t^{(i)}, \gamma^{(i)}, \mathcal{D}) &= p(\theta_x^{(i)} | b^{(i)}, \mathbf{k}_t^{(i)}, \theta_t^{(i)}, \sigma_b^{2(i)}, \gamma^{(i)}, \mathcal{D}), \\
& \quad p(\mathbf{k}_x^{(i)} | \theta_x^{(i)}, b^{(i)}, \mathbf{k}_t^{(i)}, \sigma_b^{2(i)}, \theta_t^{(i)}, \gamma^{(i)}, \mathcal{D}),
\end{aligned}
$$

which is divided into two parts:

- We sample $\theta_x$ from the conditional posterior given by

$$
\begin{aligned}
p(\theta_x | b, \mathbf{k}_t, \theta_t, \sigma_b^2, \gamma, \mathcal{D}) &\propto p(\theta_x) \int p(\mathcal{D}|b, \mathbf{k}_t, \mathbf{k}_x, \gamma) p(\mathbf{k}_x | \mathbf{k}_t, \theta_t, \theta_x) d\mathbf{k}_x, \quad (11) \\
&\propto p(\theta_x) \int \mathcal{N}(\mathcal{D}|M_t \mathbf{k}_x + b\mathbf{1}, \gamma I) \mathcal{N}(\mathbf{k}_x | 0, A_t^{-1} \otimes C_x) d\mathbf{k}_x,
\end{aligned}
$$

where the matrix $M_t$ is generated by projecting each stimulus $X_i$ onto $K_t$ and then stacking it in each row, meaning that the $i$-th row of $M_t$ is $[\text{vec}([X_i^\top K_t])]^\top$. Using the standard formula for a product of two Gaussians, we obtain the closed form conditional evidence:

$$
p(\mathcal{D}|\theta_x, \mathbf{k}_t, b) = \frac{|2\pi\Lambda_x|^{\frac{1}{2}}}{|2\pi\gamma I|^{\frac{1}{2}}|2\pi(A_t^{-1} \otimes C_x)|^{\frac{1}{2}}} \exp\left[\frac{1}{2}\boldsymbol{\mu}_x^\top \Lambda_x^{-1} \boldsymbol{\mu}_x - \frac{1}{2\gamma}(\mathbf{y} - b\mathbf{1})^T(\mathbf{y} - b\mathbf{1})\right],
$$

where the mean and covariance of conditional posterior over $\mathbf{k}_x$ given $(b, \mathbf{k}_t)$ are given by

$$
\boldsymbol{\mu}_x = \frac{1}{\gamma}\Lambda_x M_t^\top(\mathbf{y} - b\mathbf{1}), \quad \text{and} \quad \Lambda_x = (A_t \otimes C_x^{-1} + \frac{1}{\gamma}M_t^\top M_t)^{-1}. \quad (12)
$$

As in Step 1, with a uniform hyperprior on $\theta_x$, the conditional evidence is the target distribution in the MH algorithm.

- We sample $\mathbf{k}_x$ from the conditional posterior given in eq. 12.

A summary of this algorithm is given in Algorithm 1.

**Algorithm 1** fully Bayesian low-rank RF inference using blocked-Gibbs sampling

---

Given data $\mathcal{D}$, conditioned on samples for other variables, iterate the following:

1. Sample for $(b, \mathbf{k}_t, \sigma_b^2, \theta_t, \gamma)$ from the conditional evidence for $(\theta_t, \sigma_b^2, \gamma)$ (in eq. 8) and the conditional posterior over $(b, \mathbf{k}_t)$ (in eq. 10).

2. Sample for $(\mathbf{k}_x, \theta_x)$ from the conditional evidence for $\theta_x$ (in eq. 11) and the conditional posterior over $\mathbf{k}_x$ (in eq. 12).

Until convergence.

---

## 4 Approximate algorithm for fast posterior inference

Here we develop an alternative, approximate algorithm for fast posterior inference. Instead of integrating over hyperparameters, we attempt to find point estimates that maximize the conditional marginal likelihood. This resembles empirical Bayesian inference, where the hyperparameters are set by maximizing the full marginal likelihood. In our model, the evidence has no closed form; however, the conditional evidence for $(\theta_t, \sigma_b^2, \gamma)$ given $(\mathbf{k}_x, \theta_x)$ and the conditional evidence for $\theta_x$ given $(b, \mathbf{k}_t, \theta_t, \sigma_b^2, \gamma)$ are given in closed form (in eq. 8 and eq. 11). Thus, we alternate (1) maximizing the conditional evidence to set $(\theta_t, \sigma_b^2, \gamma)$ and finding the MAP estimates of $(b, \mathbf{k}_t)$, and (2) maximizing the conditional evidence to set $\theta_x$ and finding the MAP estimates of $\mathbf{k}_x$, that is,

$$\hat{\theta}_t, \hat{\gamma}, \hat{\sigma}_b^2 = \underset{\theta_t, \sigma_b^2, \gamma}{\arg\max}\, p(\mathcal{D}|\theta_t, \sigma_b^2, \gamma, \hat{\mathbf{k}}_x, \hat{\theta}_x), \tag{13}$$

$$\hat{b}, \hat{\mathbf{k}}_t = \underset{b, \mathbf{k}_t}{\arg\max}\, p(b, \mathbf{k}_t|\hat{\theta}_t, \hat{\gamma}, \hat{\sigma}_b^2, \hat{\mathbf{k}}_x, \hat{\theta}_x, \mathcal{D}), \tag{14}$$

$$\hat{\theta}_x = \underset{\theta_x}{\arg\max}\, p(\mathcal{D}|\theta_x, \hat{b}, \hat{\mathbf{k}}_t, \hat{\theta}_t, \hat{\gamma}, \hat{\sigma}_b^2), \tag{15}$$

$$\hat{\mathbf{k}}_x = \underset{\mathbf{k}_x}{\arg\max}\, p(\mathbf{k}_x|\hat{\theta}_x, \hat{b}, \hat{\mathbf{k}}_t, \hat{\theta}_t, \hat{\gamma}, \hat{\sigma}_b^2, \mathcal{D}). \tag{16}$$

The approximate algorithm works well if the conditional evidence is tightly concentrated around its maximum. Note that if the hyperparameters are fixed, the iterative updates of $(b, \mathbf{k}_t)$ and $\mathbf{k}_x$ given above amount to alternating coordinate ascent of the posterior over $(b, K)$.

## 5 Extension to Poisson likelihood

When the likelihood is non-Gaussian, blocked-Gibbs sampling is not tractable, because we do not have a closed form expression for conditional evidence. Here, we introduce a fast, approximate inference algorithm for the low-rank RF model under the LNP likelihood. The basic steps are the same as those in the approximate algorithm (Sec.4). However, we make a Gaussian approximation to the conditional posterior over $(b, \mathbf{k}_t)$ given $\mathbf{k}_x$ via the Laplace approximation. We then approximate the conditional evidence for $(\theta_t, \sigma_b^2)$ given $\mathbf{k}_x$ at the posterior mode of $(b, \mathbf{k}_t)$ given $\mathbf{k}_x$. The details are as follows.

The conditional evidence for $\theta_t$ given $\mathbf{k}_x$ is

$$p(\mathcal{D}|\theta_t, \sigma_b^2, \mathbf{k}_x, \theta_x) \propto \int \text{Poiss}(\mathbf{y}|g(M_x'\mathbf{w}_t))\mathcal{N}(\mathbf{w}_t|0, C_{\mathbf{w}_t})d\mathbf{w}_t \tag{17}$$

The integrand is proportional to the conditional posterior over $\mathbf{w}_t$ given $\mathbf{k}_x$, which we approximate to a Gaussian distribution via Laplace approximation

$$p(\mathbf{w}_t|\theta_t, \sigma_b^2, \mathbf{k}_x, \mathcal{D}) \approx \mathcal{N}(\hat{\mathbf{w}}_t, \Sigma_t), \tag{18}$$

where $\hat{\mathbf{w}}_t$ is the conditional MAP estimate of $\mathbf{w}_t$ obtained by numerically maximizing the log-conditional posterior for $\mathbf{w}_t$ (e.g., using Newton's method. See Appendix A),

$$\log p(\mathbf{w}_t|\theta_t, \sigma_b^2, \mathbf{k}_x, \mathcal{D}) = \mathbf{y}^\top \log(g(M_x'\mathbf{w}_t)) - g(M_x'\mathbf{w}_t) - \tfrac{1}{2}\mathbf{w}_t^\top C_{\mathbf{w}_t}^{-1}\mathbf{w}_t + c, \tag{19}$$

and $\Sigma_t$ is the covariance of the conditional posterior obtained by the second derivative of the log-conditional posterior around its mode $\Sigma_t^{-1} = H_t + C_{\mathbf{w}_t}^{-1}$, where the Hessian of the negative log-likelihood is denoted by $H_t = -\frac{\partial^2}{\partial \mathbf{w}_t^2} \log p(\mathcal{D}|\mathbf{w}_t, M_x')$.

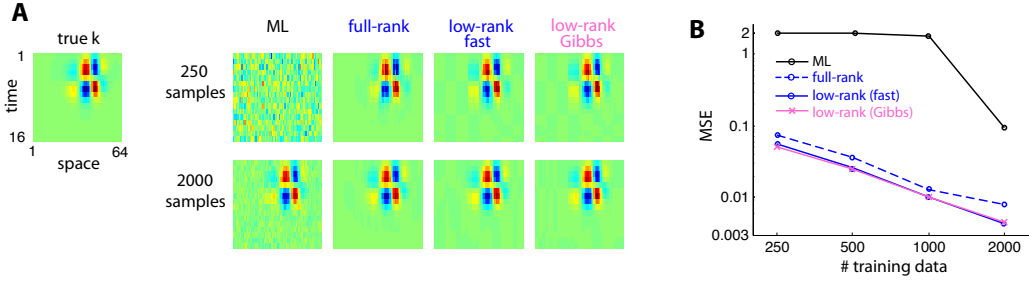

Figure 1: Simulated data. Data generated from the linear Gaussian response model with a rank-2 RF (16 by 64 pixels: 1024 parameters for full-rank model; 160 for rank-2 model). **A**. True rank-2 RF (left). Estimates obtained by ML, full-rank ALD, low-rank approximate method, and blocked-Gibbs sampling, using 250 samples (top), and using 2000 samples (bottom), respectively. **B**. Average mean squared error of the RF estimate by each method (average over 10 independent repetitions).

Under the Gaussian posterior (eq. 18), the log conditional evidence (log of eq. 17) at the posterior mode $\mathbf{w}_t = \hat{\mathbf{w}}_t$ is simply

$$\log p(\mathcal{D}|\theta_t, \sigma_b^2, \mathbf{k}_x) \quad \approx \quad \log p(\mathcal{D}|\hat{\mathbf{w}}_t, M_x') - \tfrac{1}{2}\hat{\mathbf{w}}_t^\top C_{\mathbf{w}_t}^{-1}\hat{\mathbf{w}}_t - \tfrac{1}{2}\log|C_{\mathbf{w}_t}\Sigma_t^{-1}|,$$

which we maximize to set $\theta_t$ and $\sigma_b^2$. Due to space limit, we omit the derivations for the conditional posterior for $\mathbf{k}_x$ and the conditional evidence for $\theta_x$ given $(b, \mathbf{k}_t)$. (See Appendix B).

## 6 Results

### 6.1 Simulations

We first tested the performance of the blocked-Gibbs sampling and the fast approximate algorithm on a simulated Gaussian neuron with a rank-2 RF of 16 temporal bins and 64 spatial pixels shown in Fig. 1A. We compared these methods with the maximum likelihood estimate and the full-rank ALD estimate. Fig. 1 shows that the low-rank RF estimates obtained by the blocked-Gibbs sampling and the approximate algorithm perform similarly, and achieve lower mean squared error than the full-rank RF estimates.

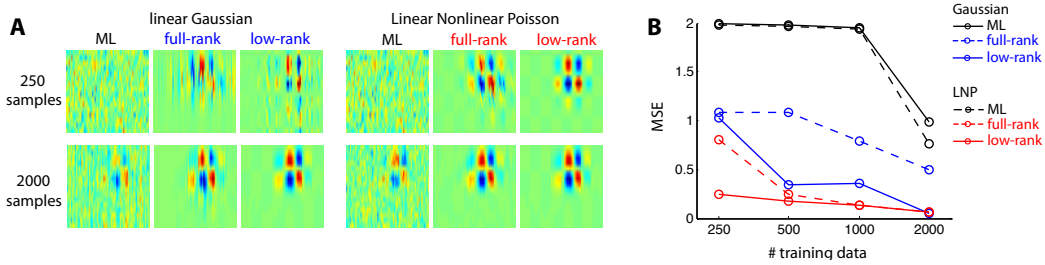

Figure 2: Simulated data. Data generated from the linear-nonlinear Poisson (LNP) response model with a rank-2 RF (shown in Fig. 1A) and "softrect" nonlinearity. **A**. Estimates obtained by ML, full-rank ALD, low-rank approximate method under the linear Gaussian model, and the methods under the LNP model, using 250 (top) and 2000 (bottom) samples, respectively. **B**. Average mean squared error of the RF estimate (from 10 independent repetitions). The low-rank RF estimates under the LNP model perform better than those under the linear Gaussian model.

We then tested the performance of the above methods on a simulated linear-nonlinear Poisson (LNP) neuron with the same RF and the softrect nonlinearity. We estimated the RF using each method under the linear Gaussian model as well as under the LNP model. Fig. 2 shows that the low-rank RF

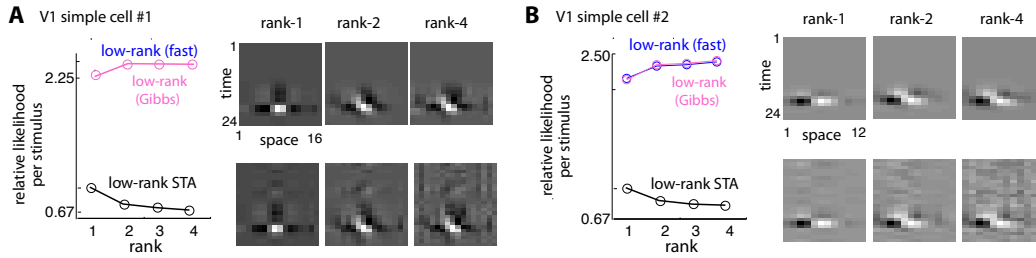

Figure 3: Comparison of low-rank RF estimates for V1 simple cells (using white noise flickering bars stimuli [16]). **A**: Relative likelihood per test stimulus (left) and low-rank RF estimates for three different ranks (right). Relative likelihood is the ratio of the test likelihood of rank-1 STA to that of other estimates. Using 1 minutes of training data, the rank-2 RF estimates obtained by the blocked-Gibbs sampling and the approximate method achieve the highest test likelihood (estimates are shown in the top row), while rank-1 STA achieves the highest test likelihood, since more noise is added to the low-rank STA as the rank increases (estimates are shown in the bottom row). Relative likelihood under full rank ALD is 2.25. **B**: Similar plot for another V1 simple cell. The rank-4 estimates obtained by the blocked-Gibbs sampling and the approximate method achieve the highest test likelihood for this cell. Relative likelihood under full rank ALD is 2.17.

estimates perform better than full-rank estimates regardless of the model, and that the low-rank RF estimates under the LNP model achieved the lowest MSE.

## 6.2 Application to neural data

We applied our methods to estimate the RFs of V1 simple cells and retinal ganglion cells (RGCs). The details of data collection are described in [16, 9]. We performed 10-fold cross-validation using 1 minute of training and 2 minutes of test data. In Fig. 3 and Fig. 4, we show the average test likelihood as a function of RF rank under the linear Gaussian model. We also show the low-rank RF estimates obtained by our methods as well as the low-rank STA. The low-rank STA (rank-p) is computed as $\hat{K}_{STA,p} = \sum_i^p d_i \mathbf{u}_i \mathbf{v}_i^\top$, where $d_i$ is the $i$-th singular value, $\mathbf{u}_i$ and $\mathbf{v}_i$ are the $i$-th left and right singular vectors, respectively. If the stimulus distribution is non-Gaussian, the low-rank STA will have larger bias than the low-rank ALD estimate.

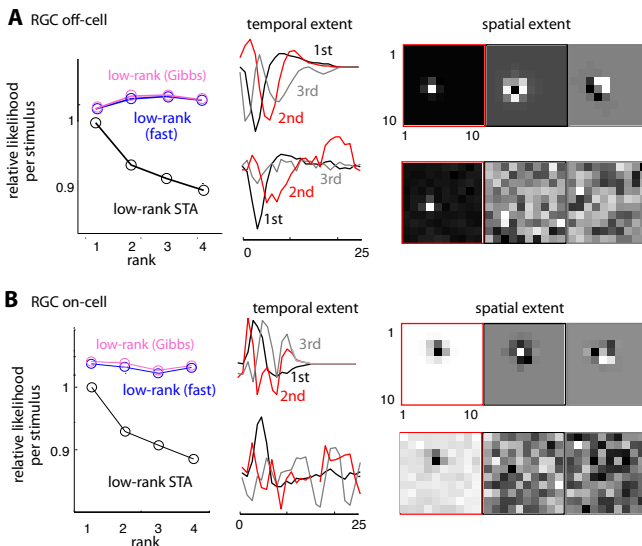

Figure 4: Comparison of low-rank RF estimates for retinal data (using binary white noise stimuli [9]). The RF consists of 10 by 10 spatial pixels and 25 temporal bins (2500 RF coefficients). **A**: Relative likelihood per test stimulus (left), top three left singular vectors (middle) and right singular vectors (right) of estimated RF for an off-RGC cell. The sampling-based RF estimate benefits from a rank-3 representation, making use of three distinct spatial and temporal components, whereas the performance of the low-rank STA degrades above rank 1. Relative likelihood under full rank ALD is 1.0146. **B**: Similar plot for on-RGC cell. Relative likelihood under full rank ALD is 1.006. Both estimates perform best with rank 1.

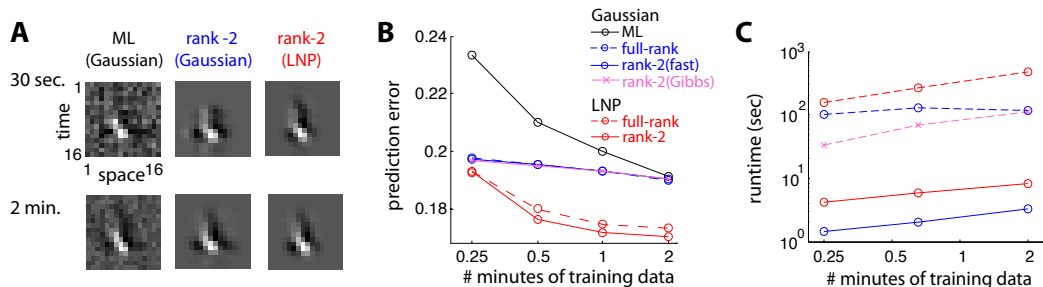

Figure 5: RF estimates for a V1 simple cell. (Data from [16]). **A**: RF estimates obtained by ML (left) and low-rank blocked-Gibbs sampling under the linear Gaussian model (middle), and low-rank approximate algorithm under the LNP model (right), for two different amounts of training data (30 sec. and 2 min.). The RF consists of 16 temporal and 16 spatial dimensions (256 RF coefficients). **B**: Average prediction (on spike count) error across 10-subset of available data. The low-rank RF estimates under the LNP model achieved the lowest prediction error among all other methods. **C**: Runtime of each method. The low-rank approximate algorithms took less than 10 sec., while the full-rank inference methods took 10 to 100 times longer.

Finally, we applied our methods to estimate the RF of a V1 simple cell with four different amounts of training data (0.25, 0.5 1, and 2 minutes) and computed the prediction error of each estimate under the linear Gaussian and the LNP models. In Fig. 5, we show the estimates using 30 sec. and 2 min. of training data. We computed the test likelihood of each estimate to set the RF rank and found that the rank-2 RF estimates achieved the highest test likelihood. In terms of the average prediction error, the low-rank RF estimates obtained by our fast approximate algorithm achieved the lowest error, while the runtime of the algorithm was significantly lower than full-rank inference methods.

# 7 Conclusion

We have described a new hierarchical model for low-rank RFs. We introduced a novel prior for low-rank matrices based on a restricted matrix normal distribution, which has the feature of preserving a marginally Gaussian prior over the regression coefficients. We used a "localized" form to define row and column covariance matrices in the matrix normal prior, which allows the model to flexibly learn smooth and sparse structure in RF spatial and temporal components. We developed two inference methods: an exact one based on MCMC with blocked-Gibbs sampling and an approximate one based on alternating evidence optimization. We applied the model to neural data using both Gaussian and Poisson noise models, and found that the Poisson (or LNP) model performed best despite the increased reliance on approximate inference. Overall, we found that low-rank estimates achieved higher prediction accuracy with significantly lower computation time compared to full-rank estimates.

We believe our localized, low-rank RF model will be especially useful in high-dimensional settings, particularly in cases where the stimulus covariance matrix does not fit in memory. In future work, we will develop fully Bayesian inference methods for low-rank RFs under the LNP noise model, which will allow us to quantify the accuracy of our approximate method. Secondly, we will examine methods for inferring the RF rank, so that the number of space-time separable components can be determined automatically from the data.

# Acknowledgments

We thank N. C. Rust and J. A. Movshon for V1 data, and E. J. Chichilnisky, J. Shlens, A. .M. Litke, and A. Sher for retinal data. This work was supported by a Sloan Research Fellowship, McKnight Scholar's Award, and NSF CAREER Award IIS-1150186.

## Footnotes

[1]In this section and Sec.4, we fix the likelihood to Gaussian (eq. 1). An extension to the Poisson likelihood model (eq. 2) will be described in Sec.5.

[2]We omit the sample index, the superscript $i$ and $(i\text{-}1)$, for notational cleanness.

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
