[Supplementary Material · NIPS2013_appendix.pdf]

## Appendix A

**Newton's method for finding a MAP estimate**

We obtain the MAP estimate $\hat{\mathbf{w}}_t$ by maximizing the log-conditional posterior

$$\Phi(\mathbf{w}_t) := \log p(\mathcal{D}|\mathbf{w}_t, M'_x) - \tfrac{1}{2}\mathbf{w}_t^T C_{\mathbf{w}}^{-1}\mathbf{w}_t + c. \tag{1}$$

The derivative expressions of $\Phi(\mathbf{w}_t)$ with respect to $\mathbf{w}_t$ are given by

$$\tfrac{\partial}{\partial \mathbf{w}_t}\Phi(\mathbf{w}_t) = \tfrac{\partial}{\partial \mathbf{w}_t}\mathcal{L}(\mathbf{w}_t) - C_{\mathbf{w}}^{-1}\mathbf{w}_t, \tag{2}$$

$$\tfrac{\partial^2}{\partial \mathbf{w}_t^2}\Phi(\mathbf{w}_t) = -H_t - C_{\mathbf{w}}^{-1}, \tag{3}$$

where $\mathcal{L}(\mathbf{w}_t) = \log p(\mathcal{D}|\mathbf{w}_t, M'_x)$, and $H_t = -\frac{\partial^2}{\partial \mathbf{k}_t^2}\mathcal{L}(\mathbf{k}_t)$. We decompose $H_t$ into three parts,

$$H_t = M'_x{}^T Z M'_x, \quad \text{where } Z = -\text{diag}\left[\frac{\mathbf{y}(gg'' - g'^2) - g^2 g''}{g^2}\right], \tag{4}$$

and $g = g(M'_x\mathbf{k}_t), g' = \frac{g}{g+1}, g'' = \frac{g}{(g+1)^2}$. The multiplication and division in eq. 4 are element by element operations.

Newton's method iterates the following:

$$\mathbf{w}_t^{new} = \mathbf{w}_t - [\tfrac{\partial^2}{\partial \mathbf{w}_t^2}\Phi(\mathbf{w}_t)]^{-1}[\tfrac{\partial}{\partial \mathbf{w}_t}\Phi(\mathbf{w}_t)],$$

$$= \mathbf{w}_t + (H_t + C_{\mathbf{w}}^{-1})^{-1}(\tfrac{\partial}{\partial \mathbf{w}_t}\mathcal{L}(\mathbf{w}_t) - C_{\mathbf{w}}^{-1}\mathbf{w}_t),$$

$$= (WW^T + C_{\mathbf{w}}^{-1})^{-1}(H_t\mathbf{w}_t + \tfrac{\partial}{\partial \mathbf{w}_t}\mathcal{L}(\mathbf{w}_t)), \text{ where } H_t = M'_x{}^T Z M_x = WW^T, \ W = M'_x{}^T Z^{\frac{1}{2}},$$

$$= C_{\mathbf{w}}(I + WW^T C_{\mathbf{w}})^{-1}(H_t\mathbf{w}_t + \tfrac{\partial}{\partial \mathbf{w}_t}\mathcal{L}(\mathbf{w}_t)),$$

$$C_{\mathbf{w}}^{-1}\mathbf{w}^{new} = (I + WW^T C_{\mathbf{w}})^{-1} b, \text{ where } b = H_t\mathbf{w}_t + \tfrac{\partial}{\partial \mathbf{w}_t}\mathcal{L}(\mathbf{w}_t)$$

$$\mathbf{w}_t^{new} = C_{\mathbf{w}} a, \text{ where } a = (I + WW^T C_{\mathbf{w}})^{-1} b$$

here, we save $a$ to avoid inverting $C_{\mathbf{w}}$ in evidence optimization. During iterations, we check if the objective, $\Phi(\mathbf{w}_t)$ is increasing. If not, we decrease the step size.

Using the notations above, the conditional log-evidence is,

$$\log p(\mathcal{D}|\theta_t, \sigma_b^2, \gamma, \mathbf{k}_x)|_{\mathbf{w}_t = \hat{\mathbf{w}}_t} \approx \log p(\mathcal{D}|\hat{\mathbf{w}}_t, M'_x) - \tfrac{1}{2}\hat{\mathbf{w}}_t^T a - \tfrac{1}{2}\log|C_{\mathbf{w}}H_t + I|,$$

## Appendix B

**Conditional posterior for $\mathbf{k}_x$ and evidence for $\theta_x$ given $(b, \mathbf{k}_t)$**

The conditional evidence for $\theta_x$ given $(b, \mathbf{k}_t)$ is

$$p(\mathcal{D}|\theta_x, b, \mathbf{k}_t) \propto \int \text{Poiss}(\mathbf{y}|g(M_t\mathbf{k}_x + b\mathbf{1}))\mathcal{N}(\mathbf{k}_x|0, A_t^{-1} \otimes C_x)d\mathbf{k}_x \tag{5}$$

The integrand is proportional to the conditional posterior over $\mathbf{k}_x$ given $(b, \mathbf{k}_t)$, which we approximate to a Gaussian distribution via Laplace approximation

$$p(\mathbf{k}_x|\theta_x, b, \mathbf{k}_t, \mathcal{D}) \approx \mathcal{N}(\hat{\mathbf{k}}_x, \Sigma_x), \tag{6}$$

where $\hat{\mathbf{k}}_x$ is the conditional MAP estimate of $\mathbf{k}_x$ obtained by numerically maximizing the log-conditional posterior for $\mathbf{k}_x$,

$$\log p(\mathbf{k}_x|\theta_x, b, \mathbf{k}_t, \mathcal{D}) = \mathbf{y}^\top \log(g(M_t\mathbf{k}_x + b\mathbf{1})) - g(M_t\mathbf{k}_x + b\mathbf{1}) - \tfrac{1}{2}\mathbf{k}_x^\top(A_t^{-1} \otimes C_x)^{-1}\mathbf{k}_x + c,$$

and $\Sigma_x$ is the covariance of the conditional posterior obtained by the second derivative of the log-conditional posterior around its mode $\Sigma_x^{-1} = H_x + (A_t^{-1} \otimes C_x)^{-1}$, where the Hessian of the negative log-likelihood is denoted by $H_x = -\frac{\partial^2}{\partial \mathbf{k}_x^2}\log p(\mathcal{D}|\mathbf{k}_x, M_t)$.

Under the Gaussian posterior, the log conditional evidence at $\mathbf{k}_x = \hat{\mathbf{k}}_x$ is simply

$$\log p(\mathcal{D}|\theta_x, b, \mathbf{k}_t)|_{\mathbf{k}_x = \hat{\mathbf{k}}_x} \approx \log p(\mathcal{D}|\mathbf{k}_x, M_t) - \tfrac{1}{2}\hat{\mathbf{k}}_x^T(A_t^{-1} \otimes C_x)^{-1}\hat{\mathbf{k}}_x - \tfrac{1}{2}\log|\Sigma_x^{-1}(A_t^{-1} \otimes C_x)|,$$

which we maximize to set $\theta_x$.