[Reviews · NeurIPS 2013]

Submitted by Assigned_Reviewer_4

This paper proposes a localized, low-rank receptive field model and develops approximate Bayesian inference methods with Gaussian and Poisson observation models.
The paper is a nice step forward in RF estimation. It’s well written, and technically sound. I enjoyed reading the paper and have only a few minor comments.

Although the paper provides clear proof-of-concept examples that show the benefits of low-rank, localized RF estimation, one area that would be especially interesting to expand on is model performance with highly non-Gaussian, natural scene stimuli. In this case low-rank STAs estimated from the whitened stimuli still have large biases that a low-rank model-based approach should resolve.

line 137 “scaler” should be “scalar”
Wording is a bit awkward at line 159 “avoids the algorithm from searching”


In response to rebuttal: all my concerns have been addressed.
Summary: This paper proposes a localized, low-rank receptive field model and develops approximate Bayesian inference methods with Gaussian and Poisson observation models.
The paper is a nice step forward in RF estimation.

Submitted by Assigned_Reviewer_5

In this paper, the authors develop a sophisticated set of Bayesian techniques to infer low-rank neural receptive fields (RFs). This is a problem of interest to many modellers and experimenters within sensory neuroscience, as identifying the transformations from stimulus spaces to output spike trains is both extremely useful for experimental protocols, and very difficult or data-intensive due to the very high dimensionality of the stimulus space. One option has been to exploit potential symmetries in stimulus space, and build low-rank approximations to RFs based on sums of space-time (e.g. Pillow et al, 2008) or frequency-time (e.g. Ahrens et al 2008) separable kernels. While these methods are useful, they have, to date, received less attention from model-development front. This has limited their widespread use.

Here, the authors extend a set of Bayesian inference techniques to low-rank RF estimation. This consists of adding hierarchical priors to the components of the reduced-rank RFs (using the ALD framework of Park & Pillow, 2012), providing full-Bayesian and fast approximate inference algorithms for the hierarchical model under a Gaussian likelihood, and fast approximate inference method for the popular linear-nonlinear-Poisson (LNP) likelihood using a Laplace approximation. The authors then demonstrate the utility (and speed) of these methods by fitting low-rank RFs to simulated and real neural data from retina and V1. Overall, this work is successful and pleasing, generally clear (constrained by the required brevity of the format), original, and likely to yield promising future results.

My only constructive comments on the paper would be to clear up a few confusions and add some minor detail:

- Lines 133-141. My understanding of this is that there are separate hyperparameters {rho, ..., Phi_f} for C_t and C_x, but this is not clear within the text. The current version is written to describe ALD for full-rank RFs, and should be amended for the low-rank cases. In the sentence "In the ALD prior..." (line 135), the localisation of C_x should be in space and spatial frequency, while that for C_t should be in time and temporal frequency. Likewise, the description of mu_s and mu_f as being centred in space-time and frequency should change (and line 140 too). Also, mu_s and mu_f as length D vectors, and Phi_s and Phi_f as DxD matrices, should be D_x / D_t and D_x by D_x / D_t by D_t respectively, with footnote 1 changed to reflect this.

- Figs 1-2: space/time labelling of axes, as in Fig. 3, would be helpful.

- The results in Figs 3-4 assume a Gaussian likelihood, yes? I am guessing this due to the use of the Gibbs sampler. This should be stated.

- Are the gains in model performance shown in Figs 3-4 the result of using the low-rank approximation, or the result of using the ALD prior? It would be helpful to quantify how much is being gained or lost by making the low-rank assumption (especially since a major contribution of this paper is to extend ALD to the low-rank case). To this end, a comparison against a full-rank ALD estimate would be of value: just demonstrating an improvement over STA is a straw man. From Fig 5B, my guess would be that low-rank ALD would perform the same as full-rank ALD for the Gaussian likelihood used in Figs 3-4. Notwithstanding this, the order-of-magnitude improvement in compute time for low-rank over full-rank remains useful.

- As always, the experimenters who would find this work most useful would also likely find it near impossible to digest. The net utility of this work depends on code being made available.
Summary: This paper provides a useful and welcome extension of Bayesian receptive field inference to the estimation of low-rank receptive fields.

Submitted by Assigned_Reviewer_6

This paper describes an elegant approach to inferring low-rank spatio-temporal receptive fields of neurons using a Bayesian formulation of reduced-rank regression. The problem is clearly specified, the implementation is nicely developed (in the form of both a sampling scheme, and a fast approximate inference, applied to both gaussian and Poisson noise models), and appears quite robust, in both simulation and on real data.

That said, I am not convinced by the magnitude of advance over existing approaches.

Most important, perhaps, are the simulation results in Figs. 1 and 2. The low-rank approaches outperform ML by a long shot, but the improvement over the full-rank ALD is small. The low-rank estimates incorporates the ALD prior; how do the low-rank estimates look without that prior? Is much of the benefit for all non-ML approaches coming from the ALD prior rather than the low-rank method per se? And for the non-simulation results, how do the low-rank results compare to the full-rank ALD estimates? Are they similar, as in the simulation? In that case, the benefit of the reduced rank approach isn't entirely clear. Indeed, even in the introduction, the notion that some form of regularization is neccessary is well articulated, but the motivation for reduced-rank versus imposing the prior (but estimating a full rank RF) is left vague. An additional concern regards the possibility of RFs that are not well described as space-time separable. How does the approach described here perform in such a setting? And how do the results compare to a full-rank, but regularized solution?

The one clear benefit from solving the reduced rank problem is speed (Figure c), though this only holds for the fast, approximate algorithm. This point could be emphasized and elaborated. Because beyond the speed improvement, although the method is nicely developed and mathematically interesting, the benefit to the RF estimation problem, compared to other regularization-based approaches, is not clear.
Summary: The authors develop a Bayesian approach for reduced rank regression to estimate low-rank neuronal spatio-temporal receptive fields. The approach is nicely developed and appears robust in simulation and in practice, but the benefit over existing approaches based on regularization (e.g. full rank ALD) is not dramatic, aside from a large reduction in speed of computation.
Author Feedback

Author rebuttal: We thank the reviewers for their careful reading of our manuscript and helpful comments for improving it. Below, we'll address a few specific comments raised in the reviews.

============
Assigned_Reviewer_4
============

> Application to natural scenes data.

This is a great idea, and thanks for the suggestion. We agree that the STA should have larger bias in such cases, and so the improvement over a low-rank STA should be even more substantial. We can easily substitute a simulated example with naturalistic stimuli in the final version.

> Typos and awkward wording: Thanks, we will fix these.

============
Assigned_Reviewer_5
============

> Separate hyperparameters {rho, ..., Phi_f} for C_t and C_x

We apologize for the confusion. Yes, these are indeed separate sets of hyperparameters governing C_t and C_x (the prior covariance for temporal and spatial components, respectively). We will make this clear in the final version.

> Figs 1-2: space/time labelling of axes

Thanks, we will add labels.

> The results in Figs 3-4 assume a Gaussian likelihood?

Yes, sorry for the omission. We will make this clear in the final version.

> Are the gains in model performance shown in Figs 3-4 the result of using the low-rank approximation, or the result of using the ALD prior? It would be helpful to quantify how much is being gained or lost by making the low-rank assumption.

Thanks for this question; we agree this is an interesting and important question. Our intuition says that both the rank constraint and the localized prior should contribute substantially to the improvement, but we will investigate this issue more closely by fitting full-rank ALD and and showing its relative performance; we
will add this to the final version. (From the simulations in Fig. 1 and 2, it's unclear whether it's having the Poisson likelihood or having data with Poisson variability that's responsible for the relative improvement of the low rank estimate; we will investigate this as well.)

> Publishing code:

Yes, we agree wholeheartedly, and definitely intend to release code.

============
Assigned_Reviewer_6
============

> Magnitude of the improvement over full-rank ALD.

We admit we're a bit surprised that full-rank ALD did so well, as we had expected a more dramatic improvement over the range of sample sizes shown in figures 1 and 2. However, we would like to draw the reviewer's attention to the first data point in Fig. 2B (performance 250 training samples). Here, the low-rank estimate achieves roughly 70% reduction in MSE compared to full-rank ALD (MSE of 0.8 vs. 0.25). So, we think that the improvement in realistic settings may be more dramatic than indicated by this pair of figures. We intend to look more closely at the low-SNR regime (larger filters, fewer number of samples), where we think the relative improvement should be greater.

> Comparison to low-rank ML estimates

We omitted the low-rank ML estimates due to the space limitations, but will add them to the revision. (The low-rank ML estimate is generally better than full-rank ML estimate, but substantially worse than low-rank estimates with the ALD prior.)

> Motivation for reduced-rank versus imposing the prior (but estimating a full rank RF) is left vague

Thanks for this comment. We will attempt to articulate this more clearly in the revision. Part of the motivation is just that (as the reviewer noted) rank constraints provide a very powerful form of regularization, since they dramatically reduce the number of coefficients. (There are also computational and memory advantages, as
the reviewer mentioned). But there is also some biological motivation for this kind of parametrization, which stems from the fact that if individual neurons have a small number of relevant timescales, then a neuron that integrates input from a large number of such neurons will also have a small number of timescales, resulting in a low-rank RF even if it pools over a large region of visual space RF. Neuroscientists have also shown particular interest in the issue of space-time separability (e.g., Adelson & Bergen 1985, showing how to construct motion detectors from a small number of space-time separable units).

> Possibility of RFs that are not well described as space-time separable.

Apologies for the confusion: none of the examples shown involved RFs that were actually space-time separable (i.e., rank-1). For example, the V1 RFs shown in Fig 3 have a clear space-time orientation, but are still well described as low-rank (optimal rank of 2 for cell #1, optimal rank of 4 for cell #2). We feel that most "standard" RFs in the early visual, auditory, or somatosensory pathways could be well parametrized as low-rank, even though they are not (for the most part) separable.

> Method is nicely developed and mathematically interesting

Thanks again for this comment. We agree, and would like to mention that we are particularly proud of the theoretical contribution of formulating a prior for low-rank RFs (and accompanying inference method) that places a marginally Gaussian prior over the RF coefficients despite the rank constraint, which puts it on equal
footing with other regularization methods that derive from Gaussian priors (e.g., ridge regression, graph Laplacian, ASD, ALD). We undertook an extensive review of the statistics literature on "reduced rank regression" and could not find any previous formulation of this idea; prior work emphasized either noninformative or Gaussian priors on the separable components (which, obviously, lead to a non-Gaussian marginal prior and posterior).